# Interactive Role of Silicon and Plant–Rhizobacteria Mitigating Abiotic Stresses: A New Approach for Sustainable Agriculture and Climate Change

**DOI:** 10.3390/plants9091055

**Published:** 2020-08-19

**Authors:** Krishan K. Verma, Xiu-Peng Song, Dong-Mei Li, Munna Singh, Vishnu D. Rajput, Mukesh Kumar Malviya, Tatiana Minkina, Rajesh Kumar Singh, Pratiksha Singh, Yang-Rui Li

**Affiliations:** 1Key Laboratory of Sugarcane Biotechnology and Genetic Improvement (Guangxi), Ministry of Agriculture and Rural Affairs/Guangxi Key Laboratory of Sugarcane Genetic Improvement/Sugarcane Research Institute, Guangxi Academy of Agricultural Sciences, Nanning 530007, China; drvermakishan@gmail.com (K.K.V.); xiupengsong@gxaas.net (X.-P.S.); lidongmei@gxaas.net (D.-M.L.); mkshmalviya@yahoo.com (M.K.M.); rajeshsingh999@gmail.com (R.K.S.); singh.pratiksha23@gmail.com (P.S.); 2Department of Botany, University of Lucknow, Lucknow 226007, India; drmunnasingh@yahoo.com; 3Academy of Biology and Biotechnology, Southern Federal University, Rostov-on-Don 344006, Russia; rajput.vishnu@gmail.com (V.D.R.); tminkina@mail.ru (T.M.)

**Keywords:** abiotic stresses, mitigation, PGPRs, plant hormones, plant growth promotion, silicon

## Abstract

Abiotic stresses are the major constraints in agricultural crop production across the globe. The use of some plant–microbe interactions are established as an environment friendly way of enhancing crop productivity, and improving plant development and tolerance to abiotic stresses by direct or indirect mechanisms. Silicon (Si) can also stimulate plant growth and mitigate environmental stresses, and it is not detrimental to plants and is devoid of environmental contamination even if applied in excess quantity. In the present review, we elaborate the interactive application of Si and plant growth promoting rhizobacteria (PGPRs) as an ecologically sound practice to increase the plant growth rate in unfavorable situations, in the presence of abiotic stresses. Experiments investigating the combined use of Si and PGPRs on plants to cope with abiotic stresses can be helpful in the future for agricultural sustainability.

## 1. Introduction

The global population is presently around 7.7 billion and may further swell in the next few years to become 10 billion by 2050 [1,2,3]. Population ballooning severely impairs the land holding capacity, particularly in Asian countries [2]. Increasing anthropogenic activities, such as the release of greenhouse gases, result in heating of the natural environment. Perturbations in climatic conditions, in turn, negatively impact food security, jeopardizing food availability and the livelihood of people [4]. Feeding such a large population is an evolving challenge for the agricultural sector and scientists [5]. Sustainable agricultural production is integral to food security; abiotic stressors, which are either the consequence of or are aggravated by climatic elements, attribute to about a 50% loss in agriculture [6]. Physical and chemical environmental factors, such as light, temperature, moisture, salinity, nutrient availability, presence of industrial and agrochemical contaminants in soil and water resources, etc., impact the growth rate and productivity. The osmotic stress caused by abiotic factors disrupts ion distribution and cell homeostasis in plants. Abiotic stresses also interact with biotic stresses, making the plant more susceptible to infestations [7,8].

Multitudes of microbial forms reside in the plant root, developing complicated microbial associations that affect plant development and production through metabolic characteristic traits and plant associations [9,10]. Variations in the morphology of plant-associated bacteria in the rhizospheric region towards the assortment of groups that are acclimated to environmental variables facilitate adaptations to environmental stresses to support better health and tolerance in plants [11]. Some of the beneficial soil bacteria, such as plant growth-promoting rhizobacteria (PGPRs), colonize the roots of plants (rhizosphere) and enhance plant performance [12] and yield [13]. PGPRs are not only linked with the roots to exert significant responses on plant growth but also have significant consequences in regulating phytopathogens [14,15]. They are also involved in improvement of the crop yield and soil profile by promoting the formation of soil aggregates and pores [16]. 

The employment of silicon (Si) in agricultural activities is another potential element for the mitigation of abiotic stresses in crop plants [2,17]. Silicon is listed as the eighth most common element in nature and the second most common element found in soil after molecular oxygen [18]. The element is found as monosilicic acid (H_4_SiO_4_) in soil; it is an uncharged molecule at pH <9 [19] and may get ionized as silicate ions, i.e., (OH)_3_SiO, in higher pH (>9). Si gets reduced in the soil from the weathering of silicate-containing minerals [20]. The dissolved content of Si varies significantly depending on the types of minerals and surrounding variables [21] as most of the soil is rich in Si. However, the availability of Si in plants is linked with soil types and seasonal environmental characteristics. Silicon has a strong affinity with O_2_ to generate silicon dioxide (SiO_2_), the most common form of Si, which constitutes around 50% of the soil mass [18,22]. The predominant form of Si in mineral soils includes SiO_2_ and primary or secondary silicate minerals having metals, such as Al (aluminosilicates) and Mg [23]. 

Silicon also exists in many irregular structures of biogenic origin, i.e., phytoliths and silica-enriched plants [24], which constitutes about 1-3% of the overall Si pool of soil [25,26]. It extends beneficial effects on plants’ fitness, performance, and productivity by mitigating abiotic and biotic stresses [17,27,28,29,30,31] along with the regulation of defense signaling pathways [32] to synergically upregulate tolerance against stress.

Both PGPRs and Si have an independent capacity to mitigate environmental stresses like salinity, water deficit, heavy metal toxicity, and nutrient deficiency [13,17,29]. Nearly 1–2% of cultivated land is affected by salinity [33], while a water deficit affects nearly 30% of the global land area. Both these stresses share several common characteristics on agriculture crop productivity [34]. Heavy metal contamination, industrial waste discharge, and pesticides are increasingly polluting the ecosystem and, thereby, plant physiology and well-being [2,35]. The present review deals with information pertaining to the mitigation of various stress factors through the application of Si and growth-promoting microbes and comparatively presents the functions by which mitigation occurs (Figure 1 and Figure 2). The information generated from the combined use of Si and PGPRs from this review could be very beneficial for environmental protection and agricultural sustainability.

## 2. Role of PGPRs and Si-Mediated Mitigation Against Stress in Plants 

The combined use of Si and PGPRs in cultivating agricultural crops has been accepted as a sustainable strategy in mitigating salinity [45] and water stress [46,47]. PGPRs may enhance biological N_2_ fixation, biosynthesis of phytohormones, and nutrient solubilization to sustain plant growth and development, with the ability to resist environmental stresses [44] (Figure 1 and Figure 2, Table 1 and Table 2). The pioneering work related with synergistic action and an improved technique for the mitigation of saline stress in *Vigna radiata* though combinatorial application of Si and PGPRs was studied [45,48]. In this section, the mechanisms and functions of PGPRs and Si-mediated stress tolerance in agricultural crop plants are discussed.

## 3. Plants’ Root Development

Si positively influences the root area development, diameter, volume, and length of the main root and root biomass in plants grown under adverse environmental variables [92,93]. It delays leaf fall, which enhances water use efficiency [20] and cell wall extensibility [2,94], synergistically extending water and nutrient absorption optimally to mitigate salinity and drought by activating specific plant hormones. Root traits also affect the plant growth rate and development directly. It is also well documented that indole-3-acetic acid-secreting PGPRs enhance root growth and exposed surfaces, benefitting water use efficiency and nutrient uptake [42,47] (Figure 1 and Figure 2, Table 1 and Table 2).

## 4. Improvement of Photosynthesis and Plant Growth

The photosynthetic machinery in plants is very sensitive to environmental stresses [17,95]. However, exogenous application of Si enhanced the photosynthetic capacity in various plants subjected to abiotic stresses [2,17,29,46,96,97]. Silicon enhanced photosynthesis, nutrient uptake, and ultimately plant growth development and biomass subjected to various stresses through increasing leaf gas exchange. Plant rhizobacteria play a crucial role in maintaining the fertility of degraded soil and improving plant performance through a wide variety of mechanisms. The mode of action of PGPRs promotes plant growth, i.e., stress tolerance in plants, easy nutrient uptake, plant growth regulators, the production of siderophores, volatile organic compounds, and protective enzymes for the protection of pest and diseases of plants [98,99,100]. Plant growth and development is influenced by a variety of stresses due to the soil environment, which is a major constraint for sustainable agricultural crop productivity.

## 5. Biosynthesis of Phytohormones

Phytohormones regulate plant morphology, affecting tolerance or avoiding abiotic stress factors [101]. By directing source and sink shifts, growth, development, and the partitioning of minerals [102], phytohormones increase the capability of plants to hold up against saline stress [102,103]. Stressed plants produce 1-aminocyclopropane-1-carboxylate (ACC), a precursor of ethylene that ultimately downregulates growth and yield. However, PGPRs containing ACC deaminase activity confer improved plant growth as water uptake continues to occur even from the deeper zones of soil [1].

Si and PGPRs were found to be associated with the enhanced adaptation of plant hormones, i.e., auxins, gibberellins, ethylene, cytokinins, and abscisic acid (ABA) were associated with saline and drought stress [55,93,104]. The production of indole acetic acid (IAA) favors the root-developing structure, root tips, and area to help plants against stress [47,105]. ABA biosynthesis may also alter gene expression induced by salt stress [106]. The downregulation of jasmonic acid and upregulation of salicylic acid in soybean was found to be associated with Si [107]. These findings indicated that the PGPRs and Si can enhance stress tolerance in plants by regulating the biosynthesis of plant hormones (Figure 1 and Figure 2, Table 1 and Table 2). There is a possible influence of Si application on the activity of PGPRs and of the soil microbial consortia.

## 6. Uptake and Translocation of Minerals 

All plants require adequate mineral nutrients for their growth and development [2]. Water deficit impairs nutrient availability and its uptake in the rhizosphere. Further, impaired nutrient translocation downregulates metabolic activities linked with growth and development [17,108]. Exogenous application of Si may balance the uptake and mobility of minerals in plants suffering from environmental stresses [20]. Sodium (Na^+^) may cause an excessive nutritional deficiency in plants [95], while the presence of Si reduces Na^+^ uptake by decreasing membrane permeability with its improved structural durability as seen in root cells [40,43]. Si protects plasma membrane integrity and permeability under salinity stress in plants [109].

The availability of Si enhances the nitrogen (N) content in plants [110], while adequate phosphorous (P) availability ensures the presence of Si in various graminaceous plants [2,18]. The application of Si may also support salt stress avoidance and the distribution of few essential minerals [111,112]. Hence, the possible influence of Si and PGPRs may upregulate growth and plant productivity under stress by balancing the uptake of nutrients [2,44]. 

## 7. Reduction of Toxic Ions

The downregulation of Na^+^ entry from the rhizosphere to the cell and the enhancement of Na^+^ efflux into the cytosol to vacuole have been proposed as a major salt tolerance mechanism [113]. However, the balance of K^+^ concentrations in the cell under salinity stress was found to be critical in maintaining proper cell functions [114]. Higher levels of Na^+^ damages cells and alters vital cellular metabolic processes, leading to the generation of reactive oxygen species (ROS) and an eventual decline in the growth rate and development [115]. Hence, plants should utilize extra cell constituents to achieve a higher level of intracellular K^+^ and a lower level of Na^+^ ions under stress [116].

Silicon may support plant resistance to salinity by minimizing the acquisition of Na^+^ through efflux via the Na^+^ - H^+^ channels, including salt overly sensitive (HvSOS_1_) and vacuolar Na^+^/H^+^ antiporter (HvNHX_1_) in the cell membrane and other membrane structures like the vacuolar membrane, respectively. The increasing K^+^ entry through K^+^ - H^+^ transport proteins like HvHAK1 was found to be beneficial [117]. Further, PGPRs are described to increase the absorption of K^+^ ions by inducing the synthesis of high-affinity ion channels, AtHKT1, in plants during salt stress with a higher K^+^/Na^+^ ratio that extends salinity tolerance [42]. PGPR strains synthesize extracellular polymers, which associate with cations like Na^+^ in order to limit its availability during salinity stress [118]. Si and PGPRs facilitate specific transport potential for K^+^ over Na^+^ and, hence, enhance the K^+^/Na^+^ balance, which could be considered as an important strategy to improve the development and fruit quality during stress [44] (Figure 2, Table 1 and Table 2). The presence of Si may decrease Na^+^ and Cl^-^ levels with an enhancement in K^+^ translocation (or an increased K^+^/Na^+^ ratio) in plants during salinity stress [2,44]. The exogenous application of Si enhances the photosynthetic characteristics in various plant species during stress [119]. It was found to reduce Na^+^ uptake by reducing the transpiration rate. Si and PGPRs, therefore, direct specific movement of K^+^ over Na^+^ and, thus, enhance the K^+^/Na^+^ ratio, which may be thought of as a major way to maintain plant development and yield against stress. 

## 8. In Vivo Accumulation of Compatible Solutes

By upregulating osmotic pressures and restricting the uptake, translocation, and distribution of water, dissolved ions (Na^+^ and Cl^-^) hinder plant morphology [116]. During stress conditions, plants need a maximum water content for development and, to some degree, a balance of the optimum water upon challenge with these stress factors through osmotic regulation [44,120]. A reduction in the relative water content of the leaf and water potential was observed in plants cultivated during drought. As per various experimental findings, salinity and water stress are associated with the upregulation of compatible solutes in plants, i.e., proline [121,122] and glycine betaine [123]. There are many reports that well-suited solutes may enhance plant tolerance to salt stress and water deficit conditions by (a) maintaining a high leaf water potential, (b) removing free radicals and maintaining the oxidation reduction redox potential, and (c) stabilizing membrane structures and proteins therein [44,102,124]. 

The improvement in the osmotic adjustment potential in response to upregulation in the osmolyte level by silicon [125] could reflect high leaf gas exchange activities with a better plant growth rate and development under stress conditions. PGPRs upregulate plant stress adaptation by enhancing the total dissolved sugar amount and storage of solutes in plants cultivated in a stressful environment [42]. Stabilizing protein–protein complexes, membranes, and osmolytes (e.g., proline, glycine betaine, amino acids, sugars (total)) reduces the negative consequences of excess ions on the antioxidant enzymatic systems [2,46] as Si enhances tolerance to a water deficit by adjusting osmolytes in various plants [39,46]. These findings indicate that Si and PGPRs may enhance the stress tolerance in plants by adjusting the osmotic potential and maintaining and/or improving the maximum leaf water content and water uptake by plants (Figure 1 and Figure 2, Table 1 and Table 2).

## 9. Response of Antioxidant Enzymes

The application of Si may mitigate oxidative damage in plants by modulating the enzymatic and non-enzymatic constituents [39,126]. Generally, the negative impact of stress on plant metabolism leads to overproduction of ROS (e.g., singlet O_2_, hydroxyl radical, H_2_O_2_ and superoxide anion), which impairs various metabolic functions, causing damage to proteins, lipids, carbohydrates, and DNA [17,20]. ROS, in turn, may also induce a damaging effect on the plasma membrane and endo-membrane systems, and disturb general metabolic processes. The main antioxidant enzymes that have been reported with enhanced activities are catalase (CAT), superoxide dismutase (SOD), peroxidase (POD), glutathione reductase, guaiacol peroxidase (GPOD), ascorbate peroxidase (APOD), and dehydroascorbate reductase [2,20,29]. Salinity and water deficit may modify the activities of antioxidant enzymes [127], while Si may reduce oxidative stress by modifying antioxidative defense enzymes [126] as it induces additional support by the formation of malondialdehyde, thereby minimizing the loss of electrolytes, malondialdehyde (MDA), and immobilization and reduction of the entry of hazardous ions [39,53]. 

PGPR-induced antioxidant enzymes are also believed to mitigate the environmental stresses (salt and water) in plants by degrading ROS in the plant roots [13]. Hence, PGPRs and silicon application may mitigate oxidative damage in various crop/plant cultivars by upregulating ROS, SOD, POD, CAT, and APOD activities along with cysteine, glutathione, ascorbic acid, and glutathione reductase [126]. It is to be noted that Si maintains cell membrane permeability and stability under abiotic stresses [2] with upgraded morphological development [44] as shown in Table 1 and Table 2. The above literature shows that the possible influence of Si and PGPRs enhances the growth rate against abiotic stresses by inducing plant synthesis of antioxidative enzymatic activities.

## 10. Improvement of Plant Water Relations

Lots of experimental studies have demonstrated that Si affects water-associated plant processes, mainly during environmental stresses [17,20,39]. Salinity- and water-stressed plants have shown a loss in water uptake, while Si application upgrades the water level and water use efficiency (WUE) in various plant cultivars [50,128,129]. The higher level of saline ions in soil results in enhanced osmotic stress, which restricts water acquisition by the plants and, eventually modulates the relative leaf water, stomatal conductance (*gs*), leaf development area expansion, and gas exchange with a reduction in the chlorophyll content and leaf greenness that affects plant development [2,113]. Improvement in the plant water balance enhances the plant dry mass [130]. Si application increases *gs* during drought to improve transpiration rates, ascribed as a positive factor to extract water from the soil [131]. Drought-challenged plants may either maintain stomatal opening to support the photosynthetic rate despite the loss of water via transpiration, or decrease the stomatal opening to retard water loss and minimize the water deficit with a loss in carboxylation [20].

In plants, the factors hindering water translocation are chiefly the roots [132]. The hydraulic conductivity of the root refers to the capacity for water acquisition, which relies on the root morphology, their permeable nature, and the applied force [133]; plants under osmotic stress primarily downregulate their root hydraulic conductivity [134]. Interestingly, Si is associated with enhanced root hydraulic conductivity through the improved expansion of plasma membrane aquaporins, which may partially confer an ability to enhance water uptake [129] and also decreases membrane damage [50]. PGPRs can upregulate water and fertilizer availability to inoculated plants, ensuring development and survival under unfavorable conditions [135]. PGPRs produce extracellular polymeric substances, such as polysaccharides, mucopolysaccharides, humic substances, and proteins [47] increasing the volume of soil macropores [136]. The above findings indicate that the Si and PGPRs enhance abiotic stress resistance in plants by upgrading the water balance.

## 11. Induced Systemic Resistance in Plants

PGPRs provoke systemic resistance and avoidance in plants. Inoculation of the stressed plants with PGPRs may trigger signaling pathways, which maximize the pathogen’s resistance to plants [137,138], and enhanced avoidance to abiotic stresses [139]. PGPRs (e.g., *Bacillus*) produce volatile organic compounds (VOCs) that may participate in the induction of systemic tolerance [140]. PGPRs and Si enhanced the ability to mitigate salt tolerance in *Vigna radiata* [45] by regulating osmolytes with a loss in lignification in the leaves, upregulating the concentration of total soluble sugar, uptake of minerals viz., K^+^, Ca^+^, Si with downregulation of Na^+^ levels similar to normal plants [45]. Si also reduces the negative effects of metal in plants by increasing the association with others. The uptake of important elements, such as zinc (Zn), iron (Fe), manganese (Mn), calcium (Ca), magnesium (Mg), P, and potassium (K), may improve their uptake [141,142]. Hence, integrated utilization of PGPRs and silicon may sustain the eco-physiological abilities of crops both under normal and stressful conditions.

## 12. PGPRs and Si Mitigating Heavy Metal Toxicity

PGPRs and Si have also been shown to mitigate the negative responses of heavy metal toxic soil [141]. Some of the known functions are shown in Figure 1 and Figure 2 and Table 1 and Table 2. Silicon has always been linked with the downregulation of the uptake and translocation of toxic ions in plants [143,144]. It is absorbed in plants through the formation of monosilicic acid, which is precipitated on the inner cell wall and lumen to chelate the ions through stimulation of the roots and exudates impairing the uptake of metal ions [141]. This minimizes the apoplastic movement of metals by reducing the bioavailability of free metals in the apoplast [145] as its accumulation in the endodermal layers of roots may reduce the mobility of metal ions [2]. Hence, through multiple functions and mechanisms, PGPRs shield the host from the phytotoxic effects of excess metal ions by altering them from bio-available to non-bioavailable pools in the soil [146]. 

By inducing physiological variations, such as the proliferation of roots and increasing the absorption rate of essential nutrients, plant hormone-producing PGPRs can mitigate toxic metal induced-stress and facilitate adaptation and tolerance [147]. Silicon and PGPRs alleviate the toxicity of metal ions in plants by enhancing the expression of genes resistant to heavy metal in plants. However, the combined application of PGPRs and Si to downregulate abiotic factors is not known to induce such changes.

## 13. PGPRs and Si Alleviating the Adverse Effects of Nutritional Deficiency

### 13.1. Macronutrients 

Silicon and PGPRs have been shown to mitigate the harmful impacts of nutritional deficiency in plants [13,148]. A few well-recognized processes by which Si and PGPRs mitigate the impact of nutrient deficient soil (micro and macro elements) in plants are depicted (Figure 1 and Figure 2, Table 1 and Table 2). The use of silicon application significantly increased the N content in plants [2]. N is a limiting nutrient for natural and agricultural eco-systems [110]. PGPRs may upregulate plant N by enhancing symbiotic and non-symbiotic N_2_ fixation and the degradation of organic N in the soil with an enhanced plant root structure by the production of a plant hormone (IAA) and enzyme (ACC deaminase) [149,150,151]. 

Phosphorus is also an essential element frequently limiting the growth and development of plants. The function of silicon in P absorption by roots was regarded as the most important finding related to Si ever studied [152]. Brenchley and Maskell [153] and Fisher [154] reported that Si fertilizer enhanced the yield of barley crops, characteristically when P fertilizer was limited. Contrary to this, when P was supplied in large quantities, Si restricted P absorption with the loss of chlorophyll, probably by downregulating the rate of transpiration [155]. PGPRs contribute considerably by three main processes of the soil P cycle, viz. sorption-desorption, dissolution-precipitation, and mineralization-immobilization. In addition, phosphate-solubilizing microbes may also enhance the accessibility of P through various processes like ACC deaminase, production of siderophores, synthesis of exopolysaccharides, production of IAA, and secretion of acidic substances [139].

Potassium (K) is another major nutrient with a crucial contribution in plant growth, development, and production [44]. K-solubilizing PGPRs convert K to bioavailable forms for plant absorption [156]. According to Mali and Aery [157], the uptake of K by the soilless system and also in the soil was found to be increased even at a reduced content of silicon dilutions through the activation of H^+^-ATPase (Table 1 and Table 2). K and Si may also enhance the availability of Ca/Mg in the rhizosphere. Si could mitigate the deficiency of K by upregulating the plant water status [64] and modulating the accessibility of K in the soil by modulating antioxidant enzyme activities [122].

### 13.2. Micronutrients 

Silicon may act as a beneficial element during nutritional imbalance. Several scientists have worked on the role of Si on micronutrient deficiency [2,158,159]. It can alleviate the impact of micronutrients on plants [158,160] by enhancing the oxidizing capacity of roots. An Si-imposed elevation in the uptake of the Fe/Mn ratio may be due to the inducing action of Si in the root-growth zone linked with an improved root length through an enhancement in cell wall extension in sorghum. The enhanced concentration of biomolecules like citrate due to the amendment of Si may lead to metal transfer from the plant root to shoot system, thus reducing micronutrient deficiency-induced characteristics [148]. Si probably contributes to balance the Fe/Mn ratio [161], increasing chlorophyll formation, providing a plausible way for the induction of growth in low-Fe plants when supplemented with Si [2,162,163]. 

Si may also alleviate Zn stress by avoiding the transfer to delicate organs in plants and increasing Zn^2+^ sorption over silicate deposits [44]. It also regulates copper (Cu) in plants cultivated in Cu-stressed conditions [164] by developing Si deposits on cell exterior, which enhances Cu-binding sites [37,148,164]. 

Iron is an important micronutrient that is lacking in plants during different conditions, viz., dryness, calcareousness, and alkalinity. IAA and siderophore produced by PGPRs may enhance Fe [105,165] with an enhanced availability of Mn in deficient plants by affecting plant growth and root development [166,167]. PGPRs can enhance the micronutrient availability to plants by lowering the soil pH and producing chelating agents [38]. Apart from this, PGPRs can maintain Cu and Zn accessibility in soil by synthesizing various components (carboxylates and phenolic compounds) and chelators (phenolics and organics acids) and also by modulating the plant growth rate and release of root exudation molecules. The substances released from the root by the activity of PGPRs develop a complex system with Mn, Fe, and others, escaping their further precipitation [38].

PGPRs may holistically manage to mitigate the unfavorable nutrient effects of soil on plant growth and development under abiotic environmental variable/stress (Figure 1 and Table 1 and Table 2). Based on the findings, it may be considered that Si and PGPRs minimize the nutrient deficit condition or excess availability in the host by regulating the amount of nutrients in agricultural crop plants. 

## 14. Conclusions and Future Prospects

In the coming decades, land degradation will be a major threat to food security. The successful use of plant-associated bacteria in contaminant removal, soil fertility, or crop protection will rely on the capability to develop strains among the previous soil-inhabiting microbes. Plant-associated bacteria play a vital role in the restoration of degraded soils through fertility enhancement by affecting nutrient cycles as well as an improvement in the soil structure. Silicon is recognized as a non-essential element needed for plant processes with its multiple beneficial impacts on growth, development, and quality productivity of plants. The use of silicon results in promotion of the physiological fitness of plants/crops for sustainable agriculture to ensure the cultivation of food crops under climatic variables, viz., abiotic stresses. The association between plants and bacteria is well recognized for its enhanced capacity to support plant growth development and improve resistance against a myriad of environmental stressors. Hence, co-cultivation of plants with bacteria, PGPRs, and Si can mitigate abiotic stresses. Their synergic effects would definitely sustain the physiological fitness of plants for improved carboxylation linked with the plant growth rate and productivity under adverse environmental variables.

Additional demonstration is inevitable to decipher the processes by which Si and PGPRs mitigate abiotic stress responses in challenged plants. More extensive information pertaining to the basic understanding regarding the combined use of PGPRs and Si would help in the development of a strategy helpful in the mitigation of environmental stresses and would probably facilitate better elucidation of plants’ responses to exposed environmental stresses. 

## Figures and Tables

**Figure 1 plants-09-01055-f001:**
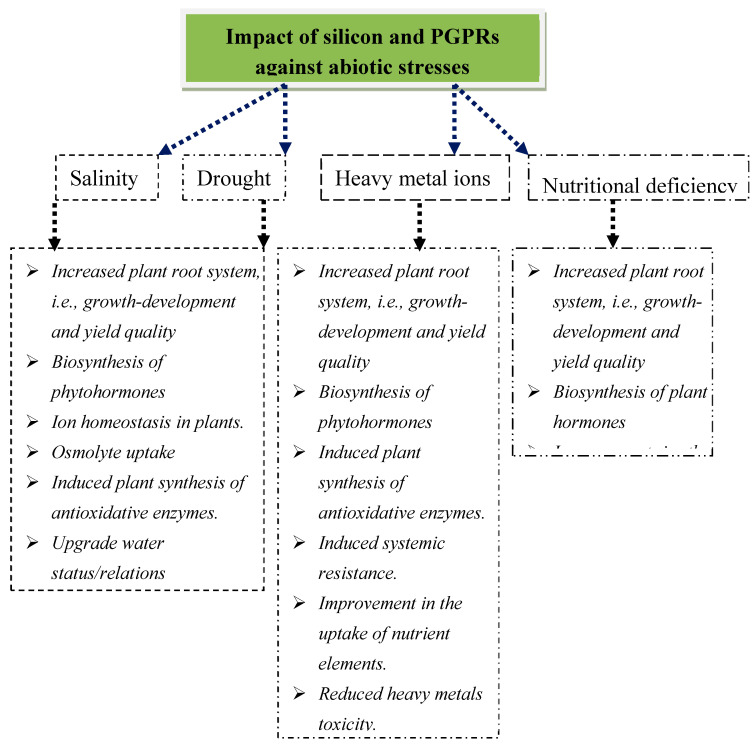
The multiple functions of silicon and plant growth promoting rhizobacteria (PGPRs) alleviate abiotic stresses in plants [17,29,36,37,38,39,40,41,42,43,44].

**Figure 2 plants-09-01055-f002:**
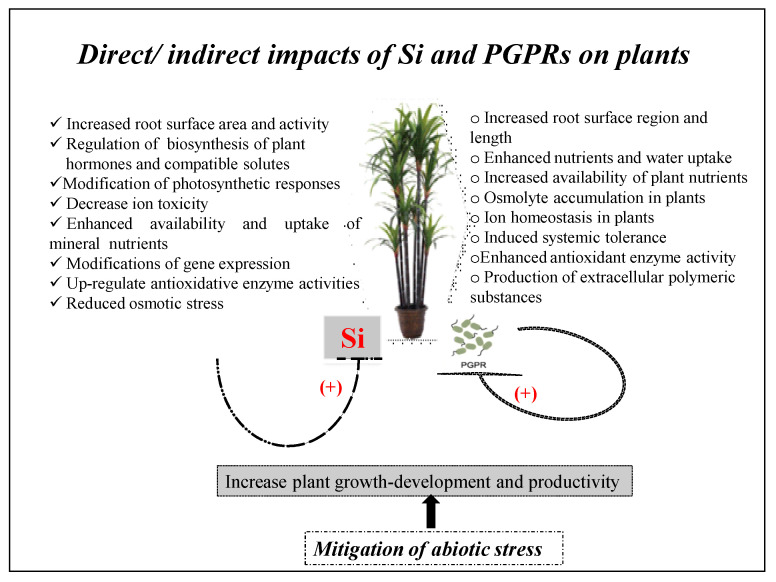
The mechanisms by which Si and PGPRs mitigate abiotic stress in plants.

**Table 1 plants-09-01055-t001:** Role of silicon on the growth, physiology, and biochemistry of plants.

Growth Condition	Plant/ Cultivars	Source of Si	Variables	Effect of Si with Stress	Reference
**Drought**	*Saccharum officinarum*	Calcium-metasilicate	Net CO_2_ assimilation	+	[17,29,30,31]
			Stomatal conductance	+	
			Transpiration rate	+	
			Photosynthesis pigments	+	
			RWC (%)	+	
			CAT	+	
			POD	+	
			SOD	+	
			Plant hormones (ABA, IAA, and GA_3_)	+	
	Strawberry (*Fragaria x ananassa ‘Camarosa*’)	Potassium silicate	Leaf area-number	no effect	[49]
			Petiole length	+	
			Electrolyte leakage (%)	no effect	
			Chlorophyll index	+	
			*Fv/Fm*	+	
			*P_N_*	+	
			E	+	
			WUE	+	
			Proline content	+	
			Shoot/ root FM/DM	+	
	*Solanum lycopersicum*	Potassium silicate	EC (%)	+	[50]
			Root hydraulic conductance	+	
			Photosynthesis	+	
			Transpiration	+	
			MDA	+	
			H_2_O_2_	+	
			CAT activity	+	
			SOD activity	+	
			Ascorbic acid	+	
			Reduced GSH activity	+	
	*Triticum aestivum* and *Zea mays*	Sodium and calcium silicate	Growth and yield traits	+	[51,52]
			Chlorophyll fluorescence	+	
			Leaf gas exchange	+	
**Salinity**	*Cucumis sativus*	Calcium silicate	Total DM	+	[53]
			Root EC (%)	+	
			Protein content	+	
			MDA	+	
			H_2_O_2_ content	+	
			CAT activity	-	
			APx activity	+	
			GPx activity	+	
			SOD activity	+	
			GR activity	+	
			DHAR activity	+	
	*Oryza sativa*	Sodium silicate	Dry mass	+	[54]
			Na^+^ content	+	
			Na+ influx and efflux	-	
			Transpiration rate	+	
			Apoplastic bypass flow	+	
	*Vigna radiata*	Potassium silicate and PGPR strains (*E. cloacae* and *B. drentensis)*	Growth traits	+	[45]
			Shoot & root fresh/ dry mass	+	
			Green pigments	+	
			Gas exchange	+	
			Salt tolerance index	+	
			Crop output	+	
	*Sorghum bicolor*	Sodium silicate	Total dry mass	+	[55]
			Photosynthetic pigments	+	
			Root Na^+^ and K^+^ content	-	
			Total polyamine content	+	
			Total ACC content	+	
	*Oryza sativa*	Silicic acid	Total no. of tillers	-	[56]
			Leaf area	-	
			Dry mass	-	
			Chlorophyll content	-	
			Proline content	+	
			SOD activity	+	
**Osmotic**	*Sorghum bicolor*	Sodium silicate	Photosynthetic responses	+	[57]
			Root hydraulic resistance	+	
			Plant dry mass	+	
	*Solanum lycopersicum*	Potassium silicate	Leaf gas exchange	+	[50]
			LWC	+	
			Root EC	+	
			MDA, H_2_O_2_	+	
			Antioxidant enzymes	+	
			Ascorbic acid content	+	
			GSH activity	+	
**Cd Toxicity**	*Oryza sativa*	Sodium silicate	Total DM	+	[58]
			H_2_O_2_ content (leaf and root)	+	
			Ascorbate content (leaf & root)	+	
			GSH content (leaf and root)	+	
			NPT content (leaf and root)	+	
**As Toxicity**	*Oryza sativa*	Sodium silicate	Photosynthesis	+	[59]
			*g*s	+	
			*g*m	+	
			*Fv/Fm*	no effect	
			Vcmax and Jmax	,,	
**Cu Toxicity**	*Spartina densiflora*	Potassium silicate	Shoot FM	,,	[60]
			Root FM and RGR	+	
			Leaf gas exchange	+	
			Chlorophyll content	+	
			Rubisco content	+	
**Mn Toxicity**	*Cucumis sativa*	Sodium silicate	Shoot fresh and dry mass	+	[61,62]
			Leaf Mn content	-	
			H_2_O_2_ and GPx activity	+	
**Al Toxicity**	*Zea mays*	Potassium silicate	Root size	+	[63]
			Root citrate and malate exudation	-	
			Root phenol exudation	-	
**K^+^ Deficiency**	*Sorghum bicolor*	Sodium silicate	Total dry wt.	+	[64,65]
			CO_2_ assimilation	+	
			Protein content	+	
			Green pigments	+	
			Leaf polyamine and arginine content	+	
			Antioxidant enzymes	+	

**Table 2 plants-09-01055-t002:** Plant–rhizobacteria-mediated plant tolerance to abiotic stresses.

Stress Condition	Plant/ Cultivar	Bacterial Inoculate	Reference
**Salinity**	*Phaseolus vulgaria*	*Azospirillum brasilense*	[66]
,,	*Vicia faba*	*Enterobacter cloacae*, *Bacillus drentensis*	[45]
,,	*Zea mays*	*Pseudomonas syringae*, *P. fluorescens*, *Enterobacter aerogenes*	[67]
,,	*Arachis hypogaea*	*P. fluorescens*	[68]
,,	*Lactuca sativa*	*Azospirillum*	[69]
,,	*Lycopersicon esculentum*	*Achromobacter piechaudii*	[70]
,,	*Triticum aestivum*	*Aeromonas hydrophila*/*caviae*, *Bacillus insolitus*, *Bacillus* spp.	[71]
,,	*Zea mays*	*Azospirillum*	[72]
,,	*Cicer arietinum, Vicia faba*	*A. brasilense*	[73]
,,	*Zea mays*	*Bacillus*	[74]
,,	*Vicia faba, Gossypium hirsutum*	*Pseudomonas*	[75,76]
**Low water**	*Lycopersicon esculentum, Capsicum annuum*	*Achromobacter piechaudii*	[77]
,,	*Triticum aestivum*	*Azospirillum*	[78]
,,	*Zea mays*	*Azospirillum brasilense*	[79]
,,	*Phaseolus vulgaris*	*Azospirillum brasilense*	[80]
**Drying soil**	*Pisum sativum*	*Variovorax paradoxus*	[81]
**Drying soil**	*Lactuca sativa*	*Bacillus*	[82]
**Osmotic stress** (PEG −45%)	*Capsicum annuum*	*Arthrobacter* spp. *Bacillus* spp.	[83]
**Osmotic stress** (PEG −20%) in dark	*Triticum aestivum*	*Azospirillum*	[84]
**Osmotic stress** (PEG −20%)	*Triticum aestivum*	*Azospirillum brasilense*	[85]
**Waterlogging**	*Lycopersicon esculentum*	*Enterobacter cloacae*, *Pseudomonas putida*	[86]
**High temperature**	*Vitis vinifera*	*Burkholderia phytofirmans*	
**High temperature**	*Solanum tuberosum*	*Burkholderia phytofirmans*	[87]
**High temperature**	*Glycine max*	*Aeromonas hydrophila*, *Serratia liquefaciens*, *Serratia proteamaculans*	[88]
**Nutrient imbalance**	*Zea mays*	*Bacillus polymyxa*, *Mycobacterium phlei*, *Pseudomonas alcaligenes*	[89]
**Iron toxicity**	*Oryza sativa*	*Bacillus subtilis*, *B. megaterium*, *Bacillus* spp.	[90,91]

Selected representation of experiments documenting a significant impact of plant–rhizobacteria on morphological, physiological, biochemical, and yield responses against environmental stresses.

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
