# Peer review of "Interactive Role of Silicon and Plant–Rhizobacteria Mitigating Abiotic Stresses: A New Approach for Sustainable Agriculture and Climate Change"

_plants, 2020, doi:10.3390/plants9091055_

Round 1

Reviewer 1 Report

1.This review article adresses a topic of interest. However, at the present stage, there are scarce experimental evidences that support a literature review on this subject.

2.In fact, the review gathers huge amount of information on the positive influence of Si on abiotic stress resistance. Plenty of high quality reviews on this have already been published and little new ideas or focus are provided here. Aslo the authors report a large array of data on the positive effects of PGPBs on abiotic stress resistance in plants. This topic has also been reviewed in extense and also nothing new here. A critical approach in both cases is missing

3. The original part of this review is the interaction of both Si and PGPBs. Unfortunately, only very few experimental data on this are currently available, so that the key subject of this review gets only scarce support from hard data. Most of the statements at the end of several paragraphs remain merely speculative; see for example line 257, or line 336, among others.

4. A point that should have been considered is the possible influence of silicon supply on the activity of PGPBs and of the soil microbial consortia.

5. Besides this fundamental criticism, the text is full of general explanations that are well known to the specialized readership of this journal. See, among many others, line 130.

6. Line 103: This subtitle is strange; better say "root area". The term canopy is ususally used for aereal parts; a canopy root  is a special arboreal adventitious root forming under certain circumstances.

In conclusion, I consider that the topic of Si- PGPBs interaction is of interest, but the scarce experimental data currently available makes it difficult to perform a specific literature review of a certain extension.  I think the topic is more appropriate for a short note or an opinion article. The simple juxtaposition of results from studies using separately Si or PGPBs is giving only rather  weak circumstancial support for a joint synergistic interaction

Author Response

Reviewer # 1

  • This review article adresses a topic of interest. However, at the present stage, there are scarce experimental evidences that support a literature review on this subject. In fact, the review gathers huge amount of information on the positive influence of Si on abiotic stress resistance. Plenty of high quality reviews on this have already been published and little new ideas or focus are provided here. Aslo the authors report a large array of data on the positive effects of PGPBs on abiotic stress resistance in plants. This topic has also been reviewed in extense and also nothing new here. A critical approach in both cases is missing

[Suggestions incorporated in the MS, as suggested]

  • The original part of this review is the interaction of both Si and PGPBs. Unfortunately, only very few experimental data on this are currently available, so that the key subject of this review gets only scarce support from hard data. Most of the statements at the end of several paragraphs remain merely speculative; see for example line 257, or line 336, among others.

[We agree with your comment. Very few research findings currently available. But nowadays so many research workers working on compound use of Si and PGPR against various stresses.  Recently, Mahmood et al. (2016) demonstrated that PGPRs and Si synergistically enhanced stress tolerance of mung bean. These researchers evaluated the effect of the two PGPR strains and two Si levels in comparison with control treatments, on mung bean under  different saline irrigations in field trial. The results of them indicated that the combined application of the B. drentensis strain with Si resulted in the greatest enhancement of mung bean physiology, growth, and yield. In another study, these researchers studied the effect of Si and the same PGPR strains on alleviating ion toxicity, regulating osmolytes accumulation, and productivity of field-grown mung bean under saline water irrigation (Mahmood et al., 2017)]

  • A point that should have been considered is the possible influence of silicon supply on the activity of PGPBs and of the soil microbial consortia.

[Corrections incorporated, as advised]

  • Besides this fundamental criticism, the text is full of general explanations that are well known to the specialized readership of this journal. See, among many others, line 130.

[Corrections incorporated, as advised]

  • Line 103: This subtitle is strange; better say "root area". The term canopy is ususally used for aereal parts; a canopy root  is a special arboreal adventitious root forming under certain circumstances.

[Correction incorporated, as suggested]

  • In conclusion, I consider that the topic of Si- PGPBs interaction is of interest, but the scarce experimental data currently available makes it difficult to perform a specific literature review of a certain extension.  I think the topic is more appropriate for a short note or an opinion article. The simple juxtaposition of results from studies using separately Si or PGPBs is giving only rather weak circumstancial support for a joint synergistic interaction.

[We agree with your nice comments but few recent studies showed that the significant results on the combined application of Si and PGPR on plants]

Reviewer 2 Report

I read carefully the paper entitled ,<< Interactive role of Silicon and plant- rhizobacteria mitigating abiotic stresses>.>First of all i would like to point out that  the topic is suitable for the journal. In my point of view the item is very interesting and with many refferings to new findings

However i have some questions for the authors

Why you haven;t mentioned anything about the apropiate concentrations of silicon and plant rhizobacteria?

You are  also reffering mostly to improvement to water relations and alleviation of nutritional deficiency but i didn;t seeanything about plant growth and photosynthetic machinery You have mentined something about chlorophyll fluoresence burt that was all.

In my point of view i need to see more references about photosynthesis and plant growth and believe me there are plenty!

Author Response

Reviewer # 2

I read carefully the paper entitled ,<< Interactive role of Silicon and plant- rhizobacteria mitigating abiotic stresses>.>First of all i would like to point out that  the topic is suitable for the journal. In my point of view the item is very interesting and with many refferings to new findings.

[Many thank you for your kind appreciation]

  • Why you haven;t mentioned anything about the apropiate concentrations of silicon and plant rhizobacteria?

[Thank you for your nice comment. Actually the concentration of Si and rhizobacteria depends on plants cultivars/ species. Si is not harmful, corrosive, and polluting to plants or environment when present in excess]

  • You are also referring mostly to improvement to water relations and alleviation of nutritional deficiency but i didn;t see anything about plant growth and photosynthetic machinery You have mentioned something about chlorophyll fluoresence burt that was all.

[Thank you for kind suggestion. Relevant literature incorporated]

Round 2

Reviewer 1 Report

This revised version makes any improvement in comparison to the first version in relation to  the interaction between Si and PGPB on stress in plants. To explore this interaction further would be of interest. However,   as the auhors do not supply further information on this, my initial consideration seems right; apparently, at present, there is too little information on this interaction to justify a review.

The influence of Si or PGPBs alone on plant stress responses have previously been reviewed in depth by several authors and this manuscript does not provide any new aspects.

This manuscript is a resubmission of an earlier submission. The following is a list of the peer review reports and author responses from that submission.